# Response of Horticultural Soil Microbiota to Different Fertilization Practices

**DOI:** 10.3390/plants9111501

**Published:** 2020-11-06

**Authors:** Iratxe Zarraonaindia, Xabier Simón Martínez-Goñi, Olaia Liñero, Marta Muñoz-Colmenero, Mikel Aguirre, David Abad, Igor Baroja-Careaga, Alberto de Diego, Jack A. Gilbert, Andone Estonba

**Affiliations:** 1Department of Genetics, Physical Anthropology and Animal Physiology, University of the Basque Country (UPV/EHU), 48940 Leioa, Bizkaia, Spain; xabier.simon@ehu.eus (X.S.M.-G.); a.martam.colmenero@gmail.com (M.M.-C.); agirremikel88@gmail.com (M.A.); dabad87@gmail.com (D.A.); igor.baroja@ehu.eus (I.B.-C.); andone.estonba@ehu.eus (A.E.); 2IKERBASQUE, Basque Foundation for Science, 48009 Bilbao, Bizkaia, Spain; 3Department of Analytical Chemistry, University of the Basque Country (UPV/EHU), 48940 Leioa, Bizkaia, Spain; olaia.linero@ehu.eus (O.L.); alberto.dediego@ehu.eus (A.d.D.); 4Department of Pediatrics and Scripps Institution of Oceanography, University of California San Diego, La Jolla, CA 92093, USA; gilbertjack@gmail.com

**Keywords:** farming system, 16S rRNA, soil prokaryotes, functional prediction

## Abstract

Environmentally friendly agricultural production necessitates manipulation of microbe–plant interactions, requiring a better understanding of how farming practices influence soil microbiota. We studied the effect of conventional and organic treatment on soil bacterial richness, composition, and predicted functional potential. 16S rRNA sequencing was applied to soils from adjacent plots receiving either a synthetic or organic fertilizer, where two crops were grown within treatment, homogenizing for differences in soil properties, crop, and climate. Conventional fertilizer was associated with a decrease in soil pH, an accumulation of Ag, Mn, As, Fe, Co, Cd, and Ni; and an enrichment of ammonia oxidizers and xenobiotic compound degraders (e.g., *Candidatus* Nitrososphaera, Nitrospira, *Bacillus, Pseudomonas*). Soils receiving organic fertilization were enriched in Ti (crop biostimulant), N, and C cycling bacteria (denitrifiers, e.g., *Azoarcus, Anaerolinea*; methylotrophs, e.g., *Methylocaldum*, *Methanosarcina*), and disease-suppression (e.g., Myxococcales). Some predicted functions, such as glutathione metabolism, were slightly, but significantly enriched after a one-time manure application, suggesting the enhancement of sulfur regulation, nitrogen-fixing, and defense of environmental stressors. The study highlights that even a single application of organic fertilization is enough to originate a rapid shift in soil prokaryotes, responding to the differential substrate availability by promoting soil health, similar to recurrent applications.

## 1. Introduction

Advances in irrigation and soil management techniques, along with the application of chemical fertilizers and pesticides introduced by the Green Revolution in farming, resulted in a substantial increase in food production over the last 50 years [1]. However, the potential of chemical fertilizers to disrupt soil health, the food chain, and ultimately human health has led to renewed interest in the consequences of their application, and has resulted in a substantial increase in the number of certified organic farms [2]. Organic systems, defined by management practices lacking the application of synthetic fertilizers and pesticides, appear to reduce the burden of xenobiotics in the food chain [3], but there is still controversy regarding the nutritional advantages of organic versus conventionally produced food. Irrespective of the impacts on food quality, the potential for organic farming to impact soil health has been expounded as a significant benefit. However, only a few studies have explored these phenomena while taking into account all the confounding variables [4,5,6].

Synthetic fertilizers can result in disrupted soil health, and may negatively impact plant growth as well as soil and plant microbial diversity and structure [7]. Therefore, the cumulative use of such compounds could lead to the loss of productivity and economic revenue [8,9]. Organic fertilizers are known to have several advantages to improve soil fertility, such as the ability to increase organic matter content in soil, improve the soil structure, enhance soil nitrogen content, enhance nutrient availability, and improve nutrient mobilization as well as increase root growth [10]. Organic practices rely upon crop rotations, crop residues, animal and/or green manure, off-farm organic wastes, mechanical cultivation, mineral bearing rocks, and aspects of biological pest control to maintain soil productivity and supply plant nutrients. It is generally assumed that greater soil microbial species’ richness promotes enhanced functional stability and thus soil health [8,11,12,13,14], but it is unknown whether short-term synthetic fertilizer application versus organic fertilizer will have a substantial impact on these ecological properties. Organic systems have previously been associated with either an increase in soil microbial richness [8] or no significant change [6,13,15,16]. Some of this variance may be explained by differences in the composition of the organic amendment, the time of harvesting, the studied time span, the rotation of crops planted, and so on. However, in general, studies suggest that short-term organic fertilizer amendment leads to a copiotrophic microbial community [5,8,15,17], while long-term application will result in a more stable community [17]. Overall, Proteobacteria and Firmicutes dominate organic farming systems in long-term field experiments, with plant growth-promoting genera also enriched (e.g., *Rhizobium, Bradyrhizobium, Burkholderia, Pseudomonas, Rhodoplanes*), while Actinobacteria and Acidobacteria predominate in conventionally managed lands [18,19].

In the present study, soil type, land use, crop rotation, crop variety, and climate/weather were all standardized in a short-term field experiment conducted on adjacent plots that share the same land use history, same soil edaphic properties, and same environmental stressors (temperature, humidity, and so on). In each plot, a synthetic chemical fertilizer and a natural fertilizer were applied, followed by planting with *Solanum lycopersicum* (tomato) and *Beta vulgaris* (Swiss chard) (hereafter referred as conventional and organic farming for simplicity). Soils were monitored for 3 months after the treatments and planting. Previously, this system was used to demonstrate that short-term organic fertilizer application did influence the accumulation of essential and non-essential elements within these crops [20,21], but overall, there was little or no clear effect of the fertilizer type on the elemental accumulation in the fruits, suggesting that the plant nutritional value was neither improved nor reduced in the short-term. Here, we employed 16S rRNA amplicon sequencing to characterize how the soil microbial community responded to these treatments and to determine to what extent the community shifts are comparable to the changes occurring in the long term. We identified biomarkers of fertilization type and correlated bacterial shifts with the accumulation of chemical elements in soil and tomato and Swiss chard roots. The proportional differences in predicted bacterial functional genes in response to the fertilizer amendment were also evaluated. Extending the duration of the experiment would help to resolve whether the microbial shifts observed here are persistent over time and determine if the treatments have an impact on soil quality in the long term, which is ultimately required for evaluating the sustainability of land-use regimes.

## 2. Results

A field experiment was conducted on adjacent plots to study the prokaryotes community diversity and composition of soils under different fertilization treatments, conventional versus organic. In each of the plots, two plant species (*S. lycopersicum* (tomato) and *B. vulgaris* (Swiss chard)) were planted in two subplots (one per crop type). Soils samples were collected over 3 months after fertilization and crop planting.

### 2.1. Soil Bacterial Community Diversity

Mean Chao1 and Shannon values were greater in the organic compared with the conventional system (Figure 1a), but these differences were not significant (analysis of variance (ANOVA), F value = 3.52, *p =* 0.0713). Similarly, there was no difference in alpha diversity with the crop (F value = 0.011, *p* = 0.917), but the date the samples were taken did show significantly different alpha diversity (F value = 3.81, *p =* 0.035). Combining “farming system” and “sampling date” resulted in a significant correlation (F value = 3.336, *p =* 0.0207). However, both conventional and organic samples showed the same overall pattern of diversity over time, with the greatest diversity in July and lowest in August (Figure 1b).

Soil pH and conductivity were not significantly correlated with alpha diversity (F = 3.342, *p* = 0.0786 and F = 0.494, *p =* 0.488, respectively).

### 2.2. Soil Microbial Composition Correlation to Experimental Factors

Community dissimilarity (beta diversity; Bray–Curtis distance) suggested a clustering of samples by ‘farming system’ (Figure 2a,b), which were shown to be significantly different (ANOSIM R = 0.541, *p* = 0.0009, Figure 2c; T-test, Figure 2d). Organic soil samples showed greater within sample Bray–Curtis values, suggesting that organic application increased the heterogeneity of soil microbial communities (Figure 2d). ‘Sampling date’ was also significantly associated with beta diversity (R = 0.15; *p* = 0.007; Figure 2c). In both farming systems, June samples (1 month after fertilizers application) harbored the most differentiated bacterial community (Figure 2b). Regarding the impact of crop variety over the soil microbial community, we found no significant differences between the soils where tomato or Swiss chard were grown (ANOSIM R = 0.066, *p* = 0.079).

Among the soil edaphic factors studied, conductivity was not associated with beta diversity nor with intervention. However, pH significantly correlated with beta diversity (Mantel r = 0.3358, *p* = 0.0010). The proportion of Bacteroidetes (specifically Bacteroidales) positively correlated with pH, as did the SHA-31 (Chloroflexi) and the *Methylocaldum* and *Cellvibrio* genera within Gammaproteobacteria (Spearman correlation, Bonferroni corrected *p* < 0.05). Genera including *Pilimelia, Rhodococcus,* and *Mycobacterium* (Actinobacteria), as well as *Rhodocyclaceae* and *Achromobacter* (Betaproteobacteria), had a significant negative correlation with pH (Appendix A).

pH was significantly greater in organic soils (Figure 3a, Appendix A) and decreased with time (June–August) for all samples, which could be influencing the time-differences observed for the microbial community. Organic soil samples collected in June (after a month of manure amendment) contained a dissimilar bacterial composition (Figure 2b and Figure 3a) and were associated with higher abundances of Ti (Figure 3a). Similarly, the abundances of soils Ag, Mn, Cd, As, Fe, Co, and Ni covaried with soil-borne bacterial community structure, but were all significantly more abundant in conventional soils (Figure 3a). According to Bioenv analysis soil pH, As and Mo were the best subset of environmental factors correlating with the soil bacterial community (Spearman correlation 0.502). While pH and As abundances were inversely correlated (Figure 3a), Mo was highest in organic soils. In addition, soil microbial composition also correlated with the accumulation of elements in the roots, where roots’ Mo and Cu concentrations were significantly associated with the bacterial community distribution of organic soils (Figure 3b).

### 2.3. Soil Prokaryotic Taxa Differences between Fertilization Practices

According to the linear discriminant analysis (LDA) effect size (LEfSe) method, 73 and 53 differentially abundant taxa were found in organic and conventional soils, respectively. Chloroflexi, Thermi, and Spirochaetes, as well as Euryarchaeota, had greater mean relative abundance in organic soil compared with conventional soil (Kruskal–Wallis test, False Discovery Rate (FDR) *p* < 0.05, Table 1; LDA score > 4, Appendix A). Within those phylogenetic groups, Anaerolineales and SBR1031 (Anaerolinea and SHA_31), members within Deinoccocales and Spirochaetes order (Spirochaetales and Spirochaetaceae families), as well as archaea belonging to Methanosarcinaceae family (*Methanosarcina* genera), were particularly enriched (Figure 4, Appendix A). Additionally, Deltaproteobacteria was substantially more abundant in organic soils (mainly because of the higher presence of Pelobacteraceae and GMD14H09). Cytophagaceae and Flammeovirgaceae (within Cytophagales), Alteromonadaceae family (Alteromonadales), Ignavibacteria, and Myxococcales orders were also augmented in organic soils (Figure 4, Appendix A). Despite having a close match in the reference database, the significantly enriched Myxococcales family Anaerobrancaceae, detected in organic soils, could not be annotated to a genus or species-level. Further inspection of the OTUs clustering within this family revealed that several of them had greater mean relative abundances in organic soils (such as OTUs 4358255, 4345857, 48487, 554552, and 564949), while other OTUs (1109458, 368942, and 4321627) were almost absent in conventional soils (Appendix A).

On the contrary, Firmicutes was significantly more abundant in conventionally treated soil samples (Kruskal–Wallis, FDR *p* < 0.05, Table 1), mainly because of the higher abundances of *Bacillus, Ammoniphillus, Paenisporosarcina,* and *Laceyella* genera (LDA scores > 5, Appendix A). Similarly, Solibacteres (in particular, *Candidatus* Solibacter, and Bryobacteraceae) and several classes within Planctomycetes (C6, Planctomycetia, and vadinHA49) were significantly enriched in conventional soils (Figure 4, Appendix A). Compared with organic soils, conventional samples also had higher relative abundances of bacteria belonging to Actinomycetales families (Micromonosporaceae, Mycobacteriaceae (*Mycobacterium*)), Nocardiaceae family (*Nocardia* and *Rhodococcus*), as well as Gemmatimonadetes (Gemmatimonadaceae) and Pseudomonadales order (*Acinetobacter* and *Pseudomonas* genera) (Appendix A). Within archaea, the relative abundances of Nitrososphaerales order (particularly *Candidatus* Nitrososphaera) were significantly higher in conventional soils.

### 2.4. Predictions of Soil Microbial Communities’ Functions

Using PICRUSt, the proportions of functional genes for each community were predicted (Appendix A) for the sequences that had a hit with the Greengenes reference OTUs at >97% identity. The weighted nearest sequenced taxon index (NSTI) score was 0.21 ± 0.015. No correlation was found between predicted functional gene richness and any of the exploratory variables tested. Similarly, no significant differences were observed in functional composition between organic and conventional soil samples (ANOSIM “farming system” R = 0.102, *p =* 0.073), nor between soil samples collected in tomato or Swiss chard plantations (ANOSIM “Crop variety” R = 0.071, *p =* 0.068). However, the ANOSIM value for the “sampling date” variable was significant (R = 0.130, *p =* 0.017). Nonetheless, LDA analysis identified five features with LDA scores higher than 2 (Log10) and significant Kruskal–Wallis and Wilcoxon rank-sum values (<0.05) when comparing soils from the two fertilization types (Figure 5). However, the histograms showed only very subtle abundance differences between farming systems (class variable) over sampling dates (subclass variable) (Appendix A). Those categories were associated with DNA repair and recombination proteins, ribosome biogenesis, homologous recombination, protein kinases, and glutathione metabolism, functions all over-represented in organic soil samples. Finally, *Bradyrhizobium, Rhodoplanes* (Rhizobiales order), and *Janthinobacter* (Burkholderiales order) had the greatest contribution to glutathione metabolism in these samples.

## 3. Discussion

While most studies agree that farming practices impact soil microbiota and the accumulation of elements within the plant, obtaining generalizable conclusions has been difficult, as results are dependent on the applied management, the composition of the organic amendments, the time of harvesting, the time span studied, the rotation of crops planted, and so on [4].

In the present study, which standardizes for differences in soil properties, crop type, and climate conditions, changes in soil microbial richness were observed over the duration of the experiment (3 months) associated with the crops’ developmental stage. Interestingly, microbial richness was greatest in July (2 months after fertilizers application) and lowest in August for both conventional and organic soils, suggesting that while nutrient supplies start to decrease, there is a decrease in richness. In addition, the combined effect of ‘sampling date’ and ‘farming system’ significantly correlated with bacterial richness, with the values being higher in organic soils. While the higher richness in organic systems might be in part due to the introduction of microorganisms present in the manure into the soil (represented mainly by members within Firmicutes (*Clostridia*), Bacteroidetes, and Chloroflexi) [22], previous data suggest that organic farming systems promote habitat diversification, favoring a more heterogeneous species distribution [18,19], or by stimulating the growth of copiotrophic microorganisms [8,17]. In conventional soils, lower diversity might be expected because of the elimination or growth inhibition of particular bacteria in response to chemical compounds coming from pesticides/fungicides [16,23,24]. Predicted functional diversity, however, did not change according to the farming system.

pH is known to influence microbial composition [25,26,27,28,29] as well as the mobility of heavy metals, influencing micronutrients’ uptake [30]. In the present study, while pH was a determinant factor explaining the bacterial community structure found in the soil samples, it did not correlate with the alpha diversity estimate. Despite both plots (organic and conventional) starting at the same pH values, conventional fertilizer resulted in a reduced pH over time, consistent with the observed impact of chemical fertilizers in longer-term experiments [23,24]. pH variation influences Proteobacteria, Actinobacteria, and Acidobacteria abundances [25,29,31]. For example, Lauber et al. [28] found that the relative abundance of Acidobacteria decreased with pH, while Actinobacteria and Bacteroidetes positively correlated with soil pH. In our study, Bacteroidetes positively correlated with pH, while Actinobacteria was negatively correlated. The different trends observed in this study might result from the soil having only been fertilized once, which would select for copiotrophic taxa, compared with other studies where multiple recurrent fertilizations were applied over years. In addition, titanium (Ti) abundance was positively associated with the community structure found in organic soils, and negatively associated with that of conventional soils. In contrast, the abundances of Ag, Mn, As, Fe, Co, Cd, and Ni were mainly associated with conventional soils. Titanium is considered to be a beneficial element for plant growth, improving crop performance through stimulating the activity of certain enzymes, enhancing chlorophyll content and photosynthesis, promoting nutrient uptake, strengthening stress tolerance, and improving crop yield and quality [32]. In agreement, organic soils in the present study shifted towards the enrichment of taxa involved in nutrient cycling as well as in disease suppression. When studying the accumulation of chemical elements within the plants, Liñero et al. [20] documented a differential accumulation according to the fertilization practice. Higher concentrations of Mn, Co, Na, Mg, Cd, and Tl were observed in conventionally grown tomato plants, while Mo, Cu, Zn, K, and Ba abundances were higher in the organically grown ones. Similarly, Swiss chards under organic amendment were more concentrated in Ba and some nutrients (Na, K, Mn, and Mo) [21]. Interestingly, the soil bacterial community of the present study was a good predictor of Mo and Cu accumulation in organic tomato and Swiss chard roots, thus suggesting that those elements’ absorption is favored, in part, by means of the soil microbial activity. For instance, because of the spraying of copper sulfate on plant aerial parts, a higher accumulation of Cu might be expected in organic roots. Besides, its higher concentration in organic agricultural practices has been previously associated with a higher presence of arbuscular mycorrhizal fungi (AMF) [33,34], and their synergistic interactions with several bacteria, such as species belonging to Rhizobiales and Methylococcales, are already well known [35].

The soil’s bacterial community composition was significantly influenced by the management practice in the 3 months of the experiment, as observed in previous long-term studies [6,17,24]. However, the experiment conducted should be extended over time to assess whether the microbial shifts associated with the farming system and the differential uptake of elements by the crops under study persist, in order to evaluate the farming system’s impact on the quality and health of soil, and hence the sustenance of the system. In any case, in the present study, similar to Lupatini et al. [5], but in contrast to Wang et al. [36], organic samples tended to have greater beta diversity compared with conventional samples, suggesting a greater heterogeneity in the microbial composition of organic soils. Hartmann et al. [17] found that ~10% of bacterial and fungal OTUs were specific to the farming system (conventional vs. organic) and Lupatini et al. [5] reported that Proteobacteria and Acidobacteria were highly sensitive to the farming practice. In the current study, the abundances of most common bacterial phyla were not statistically different between farming systems, but few phyla were significantly associated with each of the farming systems. For instance, Chloroflexi, Thermi, Spirochaetes, and Euryarchaeota had greater mean relative abundance in organic soils, while a higher abundance of Firmicutes was observed in conventional soils. Within Chloroflexi, members of the class Anaerolineae (SBR1031, SHA31) were enriched, which are known for their role in nitrogen cycling [6] and have been previously identified as a highly represented bacterial group in manure [22]. Several other denitrifiers were also augmented compared with conventional soils, including genera *Azoarcus* and *Thauera, Parvibaculum,* and *Saccharomonospora,* while nitrifiers were depleted. Ding et al. [37] observed a similar result when studying microbial community changes in a long-term organic greenhouse farming, where the relative abundances of ammonia oxidizing microorganisms and anaerobic ammonium oxidation bacteria decreased in the organic soil. Furthermore, an increase in methylotrophic bacteria, for instance, those belonging to families Methylophilaceae (particularly *Methylobacillus*) and Methylococcaceae (*Methylocaldum*), as well as the archaea *Methanosarcina,* was observed, likely associated with their capability to metabolize methane and its derivative compounds that accumulated after the decomposition of the introduced organic matter in organic farming. Taxa belonging to Myxoccocales, Alteromonodaceae, and various Rhizobiales OTUs, known to contribute to general nutrient cycling (C, N, S, and P), were also more abundant in organic soils, similar to previous reports [36]. Members of Ignavibacteria order, suggested to be involved in the degradation of organic matter, were higher in the soil under the organic farming, similar to the results observed in longer-term field and greenhouse studies [37,38]. However, other groups known to be capable of degrading various complex organic materials coming from manure or compost, such as several genera within Firmicutes phyla [17], did not respond in that direction. While it could be speculated that they might need recurrent organic amendments to respond, a 12-year greenhouse study suggested that Firmicutes were the least affected phyla by farming system [37].

Interestingly, previous studies report that organic farming systems tend to increase the abundance of microbial taxa associated with plant health promotion [18,19,39]. Several members of Firmicutes and Actinobacteria have been associated with disease suppression and have been reported to be augmented in organic farming [36]. However, this was not the case in this study after a one-time manure application, where, for instance, higher abundances of *Bacillus, Nocardia*, *Mycobacterium,* and *Rhodococcus* were observed in the conventional soils. Besides, the plant growth-promoting Myxococcales was ~3.5 times more abundant in organic compared with conventional soils. They are considered to be micropredators that can produce secondary metabolites with antibacterial and antifungal activities, killing other microorganisms [40], and as such, have been suggested to likely play a key role as potential disease-suppressing bacteria in organic farming soils. Interestingly, Myxococcales, unlike the mentioned Firmicutes and Actinobacteria members, have been consistently found to be enriched in organic soils in both short- (this study) and long-term experiments [24,36], thus suggesting a rapid and lasting response of this bacteria to the organic amendment. Thus, these organisms’ population distribution and functional genes deserve further investigation.

In good agreement with other studies, Solibacteres, mainly *Candidatus* Solibacter, previously suggested to be adapted to nutrient-limited environments [17], was associated with the conventional farming system. Interestingly, taxa capable of degrading xenobiotic compounds (e.g., *Pseudomonas*, Paenibacillaceae, *Bacillus*) were also enriched [15]. The nitrification process was enhanced, as ammonia oxidizing bacteria (Nitrospira and *Mycobacterium*) and archaea (*Candidatus* Nitrososphaera) were particularly induced after applying the conventional fertilization, and that response was consistent over the 3 months of the experiment, similar to longer-term studies [38]. Thaumarchaeota archaea’s enrichment has been previously observed after a long-term application of organic fertilizers [41] and in several long-term fertilization experiments with inorganic N treatment [42,43]. In addition, several denitrifying bacteria responded to the chemical fertilization, such as *Gemmatimonas* (Gemmatimonadetes), *Pseudomonas* (Gammaproteobacteria), *Achromobacter* (Betaproteobacteria), *Nocardia*, and *Rhodococcus* (Actinomycetales).

In this study, functional profiles were more resistant to intervention than community composition. This agrees with the conclusions of Pan et al. [7], who proposed that the coexistence of organisms with overlapping ecological functions confers functional stability. Fierer et al. [11] found that, under high concentrations of inorganic nitrogen, the relative abundance of the DNA/RNA replication, electron transport, and protein metabolism functions increase. Likewise, Carbonetto et al. [44] evidenced that the relative abundances of intracellular trafficking, secretion and vesicular transport, energy production and conversion, and amino acid transport and metabolism were greater in soils under conventional farming system than in uncultivated soils, consistent with a copiotrophic strategy. Ding et al. [37] reported changes in functional groups associated with nitrogen cycling when conducting a metagenomic analysis, observing the greatest effect for functional groups associated with aerobic ammonia oxidation, nitrite reduction, anaerobic ammonia oxidation, and nitrate reduction. Chen et al. [6], besides, found no significant differences in functional genes, predicted from 16S RNA using PICRUSt, involved in denitrification (nirK and nosZ), nitrification (hao), and N-fixation (nifH) when analyzing soils managed over 18 years that included organic and conventional farming. Similarly, in the present study, when evaluating PICRUSt predicted functions, the Kruskal–Wallis test did not detect any differentially abundant functions between the conventional and organic soil samples, which included those N transformation functions. However, according to LDA, organic soils had greater predicted abundances of glutathione metabolism, which plays an important role in the defense of microorganisms and plants against environmental stresses. In addition, it is also involved in the regulation of sulfur nutrition and plays a key role in the nitrogen-fixing symbiotic interaction [45]. However, the functional results reported here are based on predictions obtained from the 16S rRNA gene, which resulted in NSTI values that were moderately high [46], as expected for phylogenetically diverse samples such as soil [12], suggesting that those predictions must be interpreted with caution. In addition, functional differences might have been hidden, as de novo OTUs were eliminated for the analysis to conduct PICRUSt predictions.

## 4. Materials and Methods

### 4.1. Field Experimental Design

The study was carried out in two adjacent plots of 25 m^2^ separated by 35 m, located in Beotegi, a rural area of the Basque Country (43°5.370′ N; 3°4.590′ W, Spain), at 370 m above sea level.

Soils from both plots were collected in February 2013 for physicochemical analysis, prior to the amendment of fertilizers and plant plantation. The physicochemical analysis conducted in these samples has been published in Liñero et al. [20,21]. The data obtained from the soil characterization ensured there was no significant difference among the edaphic properties of the plots intended for conventional or organic practices prior fertilizer amendment and plant plantation (Kruskal–Wallis and ANOVA tests > 0.05).

In the plot intended for conventional practice, 0.25 kg·m^−2^ of a synthetic chemical fertilizer (NPK 15.15.15 (15); Fertiberia, S.A., Burgos, Spain) was applied. Therefore, the conventional plot was supplied with a dose of 938 g N, P_2_O_5_, and K_2_O. Twenty-five days later, in June 2013, twenty-five seedlings of Swiss chard (*Beta vulgaris* subsp. *adanensis*) and tomato (*Solanum lycopersicum*) were transplanted into two sub-plots (one per crop type). Likewise, a phytosanitary treatment composed of a liquid mixture of an insecticide (Epik 20SG; Sipcam Jardín S.L., Valencia, Spain) and a fungicide (Galben M.; Sipcam Jardín S.L., Valencia, Spain) were applied twice (200 mL·m^−2^ in total), at 7 and 36 days after plantation.

In the plot intended for organic practice, however, a natural fertilizer (natural horse manure, Abonos Naturales Hermanos Aguado, S.L., Toledo, Spain; product approved and certified by CAEE as ecological product; C qualification) was supplied with two applications 10 and 2 days before planting Swiss chard and tomato seeds. Altogether, 6.48 kg·m^−2^ was applied, which corresponds to a dose of 7452 g of total N, 648 g of P_2_O_5_, and 1458 g of K_2_O. Twenty-five days later, Swiss chard (*Beta vulgaris* subsp. *adanensis*) and tomato (*Solanum lycopersicum*) were transplanted within the plot into two sub-plots (one per crop type). As a natural repellent to avoid pests and insect attacks, protective plants (*Tagetes patula*) were planted in the periphery of the organic plot. Additionally, a total of 0.67 g m^−2^ of powdered copper sulfate (Desarrollo Químico Industrial, S.A., Valencia, Spain; product approved and certified by SHC for organic farming) was sprinkled on the aerial parts of the tomato plants 7 and 36 days after plantation to ensure the absence of typical fungi found in tomato cultivars, such as mildew.

### 4.2. Soil Sampling

Three sampling campaigns were performed in the organic and conventional plots, collecting samples from soils associated with tomato and Swiss chard plants on 26 June, 15 July, and 26 August (1, 2, and 3 months after fertilization). Soil samples (0–10 cm layer) were collected in triplicate in each day and plot, using a soil corer, and were preserved in pre-labelled zip bags and kept in portable coolers at low temperature during their transportation to the laboratory. Once in the laboratory, samples were stored in the darkness at −20 °C until their processing. In total, 36 soil samples were gathered: 18 conventional samples (9 from Swiss chards and 9 from tomato plants) and 18 organic samples (9 from Swiss chards and 9 from tomato plants, respectively).

For soil property analysis, soil was dried and pH and conductivity were measured in a soil/water ratio of 1:2.5.

The soil chemical elements including the following macronutrients (Mg, K, and Ca), micronutrients (Na, Mn, Fe, Co, Cu, Zn, Ni, Mo, and Se), and nonessential elements (Sr, Ag, Ba, W, Al, Ti, V, Cr, As, Sn, Sb, Cd, Tl, Hg, and Pb) were measured in Liñero et al. [20,21] and evaluated in this study.

### 4.3. DNA Extraction, PCR Amplification, and 16S rDNA Sequencing

Here, 0.25 g of each soil sample was used as an input for DNA extraction using the DNA PowerSoil kit (MoBio laboratories, Inc., Carlsbad, CA, USA). PCR amplification of the V4 hyper-variable region of the 16S rRNA gene was performed using the 515F/806R barcoded primers and following the Earth Microbiome Project protocols [47]. Each 25 µL PCR reaction contained 12 µL of MoBio PCR Water (Certified DNA-Free), 10 µL 5 µM HotMasterMix (5 Prime), 1 µL of Forward Primer (5 µM concentration, 200 pM final), 1 µL Golay Barcode Tagged Reverse Primer (5 µM concentration, 200 pM final), and 1 µL of template DNA. The conditions used for the amplification were a hot start of 94 °C for 3 min, followed by 35 cycles of 94 °C for 45 s, 50 °C for 60 s, 72 °C for 90 s, and a final elongation at 72 °C for 10 min. PCR amplifications were pooled into a single tube in equimolar concentrations. The pool was then cleaned using the UltraClean^®^ PCR Clean-Up Kit (MoBio laboratories, Inc, Carlsbad, CA, USA), and quantified with Qubit (Invitrogen). Finally, the molarity of the pool was determined and diluted to 2 nM, and denatured for sequencing on the Illumina MiSeq platform (150bp × 2 pair-end) at Argonne National Laboratory (USA).

### 4.4. Sequence Data Processing and Statistical Analysis

Sequences processing was performed using QIIME v. 1.9.1 [48]. First, a demultiplexing process was carried out to perform a quality-trimming step and to link each sequence to the sample it came from using barcodes information. An open reference OTU picking was carried out, whereby OTUs were clustered against GreenGenes 13_5 reference sequences, and reads that failed to hit the reference were subsequently clustered de novo by their 97% similarity level using uclust [49]. OTU taxonomy was assigned using the RDP classifier [50] against the GreenGenes database (97% similarity). Sequences were aligned using PyNAST [51] and a final OTU table was created, excluding unaligned sequences and singletons.

To avoid biases related to using different sequencing depths per sample in the analysis, the OTU table was rarefied to a number of sequences in which the full range of microbial diversity was investigated. Thus, the first rarefaction plots were created with the full number of sequences yielded in the sequencing and the resultant rarefaction curves were used to choose the rarefaction depth.

Following removal of samples with low read depth or obvious contamination, 29 samples remained, rarefied to 32,995 sequences per sample, which clustered into 8578–11327 OTUs/sample. OTUs were collapsed at different taxonomic levels and bar plots were created to obtain an overview of the overall composition of the microbial community in the samples. Species richness and Shannon diversity indexes were calculated for all soil samples in Phyloseq, R package [52]. In addition, the correlation between the alpha diversity measurements and the variables tested in this study—“farming practice”, “sampling date”, “crop variety”, “pH”, and “conductivity”—were evaluated through ANOVA test in R, vegan package [53].

Principal coordinates analysis (PCoA) and canonical correspondence analysis (CCA) were constructed to visualize differences between microbial communities based on Bray–Curtis phylogenetic dissimilarity using QIIME and Phyloseq package functions. In order to test whether there were significant community differences according to the exploratory variables, ANOSIM tests based on Bray–Curtis dissimilarity were performed, through 1000 permutations per test. The influence of the exploratory factors—“farming system”, “sampling date”, “crop variety”, “pH”, and “conductivity”—and the soil macronutrients (Mg, K, and Ca), micronutrients (Na, Mn, Fe, Co, Cu, Zn, Ni, Mo, and Se), and nonessential elements (Sr, Ag, Ba, W, Al, Ti, V, Cr, As, Sn, Sb, Cd, Tl, Hg, and Pb) measured from the same samples in Liñero et al. [20,21] were also tested by CCA using vegan package. The significance of the overall model used to create the CCA was obtained through ANOVA test, followed by stepwise ordination, obtaining the significance *p*-value for each of the variables included in the model. Furthermore, Mantel test was used to evaluate the correlation between soil edaphic properties (pH and conductivity) and community distances. Spearman rank test was used to identify the genera significantly correlated with the edaphic factors under study. To evaluate whether pH and conductivity values differed by farming system, boxplots function was used and a Mann–Whitney U test was estimated. The relationships of soil bacteria composition and soil and root chemical element abundances were evaluated through non-metric multidimensional scaling plot (NMDS) ordination and envfit procedure in Vegan R package [53]. BIOENV method was used to perform permutation tests on environmental variables for determining which variables produce the highest correlation with the bacterial composition, calculating Spearman rank correlations between Euclidean distance matrices (environmental data) and Bray–Curtis distances (biological data).

The mean beta diversity distances of samples by farming system, organic and conventional, were visualized and its significance was evaluated by performing two-sample *t*-tests with 999 Monte Carlo permutations. Additionally, Kruskal–Wallis test was evaluated to check for OTUs whose abundances significantly differed between “farming practice”, using the corrected FDR *p* value (<0.05) as the significance criteria, and a heatmap tree was constructed with the significant OTUs. In addition, significant taxonomic differences between farming systems were also tested using lineal discriminant analysis effect size with LEfSe tool in Galaxy [54]. LEfSe performs non-parametric factorial Kruskal–Wallis sum-rank test (0.05) to detect taxa with significant differential abundances followed by Wilcoxon rank-sum test (0.05) for evaluating consistency and a final step of LDA to estimate the effect size of each differentially abundant feature. Significant taxa were used to generate taxonomic cladograms and abundance histograms to illustrate biomarkers abundance differences between farming practices.

Microbial communities’ functions were predicted from the 16S rRNA data using PICRUSt v 1.1.1 software [46]. First, a close reference OTU table was created and those OTUs 16S rRNA copies were normalized by dividing their abundances by known or predicted 16S copy number abundances. Functions were assigned based on KEGG database [55]. The functions table was then used for analyzing alpha diversity differences among exploratory variables, and to investigate functional profile differences applying ANOSIM test. In addition, the predicted function table was used for biomarker identification in LEfSe, where LDA score and feature histograms were created using “farming system” as class and “sampling date” as subclass. In addition, the OTUs’ contribution to the functional features that showed significant differences according to LDA were inspected using PICRUSt’s metagenome_contributions.py.

## 5. Conclusions

Soil microbiota plays a crucial role in maintaining soil health, and as such, many studies have attempted to determine the differences in microbial community composition, structure, and function between different farming systems. However, most studies have lacked effective control between soil and plant variables, and comparability between studies is virtually impossible because of the variability in crop varieties, soil tillage, fertilization, and plant protection strategies and dosage. The standardized experimental design applied in this study removed much of the bias associated with previous studies. As we found similar impacts of farming system in our short-term study when compared with longer-duration studies, it can be inferred that even short-term organic system adoption has a considerable impact on soil microbiology and soil health. However, further research is needed to confirm these results in the long term, which altogether would allow elucidating the connection between the observed changes and plant productivity, disease resistance, and stress resilience.

## Figures and Tables

**Figure 1 plants-09-01501-f001:**
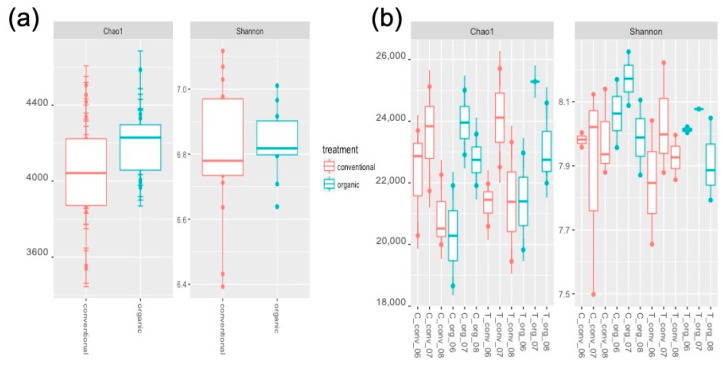
Effect of farming system on bacterial community alpha-diversity. (**a**) Chao and Shannon diversity estimates in organic and conventional soil. (**b**) Chao and Shannon diversity estimates in organic and conventional soils, by crop variety and sampling date. The center-line of the boxplots show the medians, and the bottom and upper limits indicate the 25 and 75th percentiles, respectively. Crop variety: C (Swiss chard), T (tomato). Sampling date: 06 (June), 07 (July), 08 (August). Farming system: Conv. (conventional); Org. (organic).

**Figure 2 plants-09-01501-f002:**
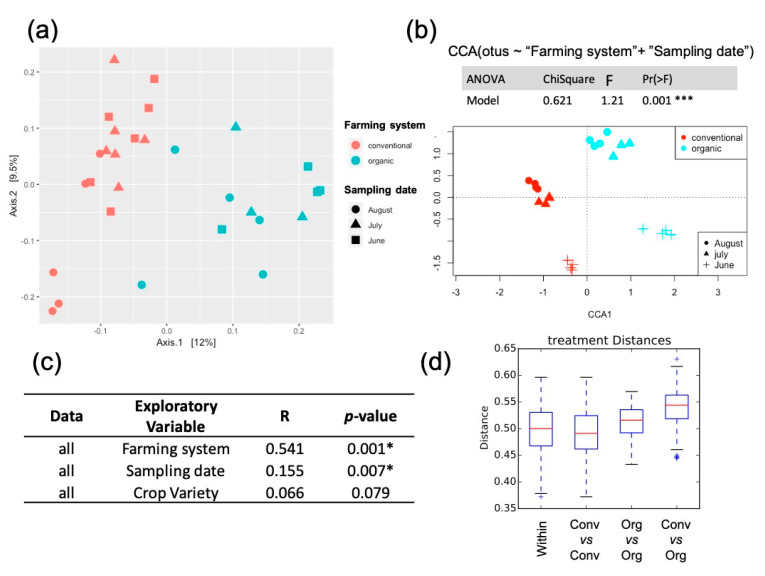
Effect of exploratory variables on bacterial community beta diversity. (**a**) Principal coordinate analysis (PCoA) among soil samples based on Bray–Curtis distance. Samples are colored by “farming system” (conventional vs. organic), and shapes represent “sampling date”. (**b**) Canonical correspondence analysis (CCA) constrained by “farming system” and “sampling date”, showing the correlation between those factors with microbial communities. A forward stepwise selection including all the categorical factors studied (farming system, sampling date, crop variety), soil edaphic factors measured (pH and conductivity), and the soil chemical elements (macronutrients, micronutrients, and non-essential elements) evaluated in Liñero et al. [20,21] were originally included in the tested model and the significant factors (farming system and sampling date) were used to construct the ordination. The analysis of variance (ANOVA) results show the significance for the best predictive CCA model (Chisq: chi-square test; F: F value; Pr: probability). *** = *p* ≤ 0.001. (**c**) ANOSIM values of category effects on microbial diversity patterns. * = *p* ≤ 0.05. (**d**) Beta diversity boxplots for organic and conventional soils. The center-line of the boxplots show the medians, and the bottom and upper limits indicate the 25 and 75th percentiles, respectively. * indicates significant two-sided Student’s *t*-test values.

**Figure 3 plants-09-01501-f003:**
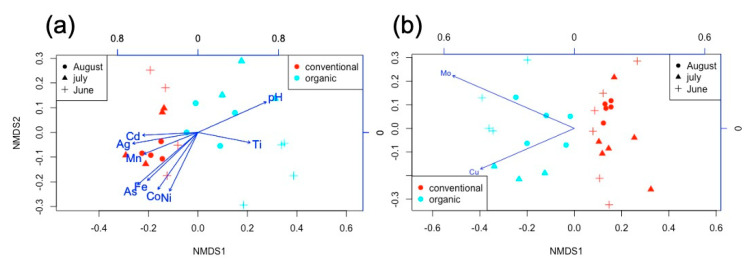
Non-metric multidimensional scaling plot (NMDS) of soil bacterial community using OTU(Operational Taxonomic Unit)-based Bray–Curtis dissimilarities distances overlaying chemical elements from (**a**) soil and (**b**) tomato and Swiss chard roots (element abundances obtained from Liñero et al. [20,21]). Only elements with *p* < 0.05 were plotted and the arrow length is proportional to the strength of correlation.

**Figure 4 plants-09-01501-f004:**
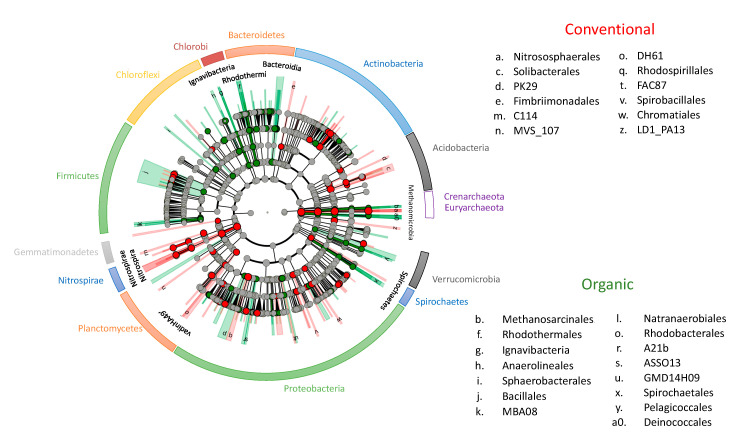
Cladogram representation of statistically and biologically consistent differences between organic and conventional soils. Taxonomic groups enriched (linear discriminant analysis (LDA) effect size (LefSe) analysis based on Kruskal–Wallis *p* < 0.05, and LDA scores Log10 > 2) in the conventional soils are represented with red dots, while green dots are associated with organic soil. The legend indicates the differentially abundant orders per farming system. Genera level information and the LDA score values for each group are available in Appendix A (organic soils biomarkers) and Appendix A (conventional soils biomarkers).

**Figure 5 plants-09-01501-f005:**
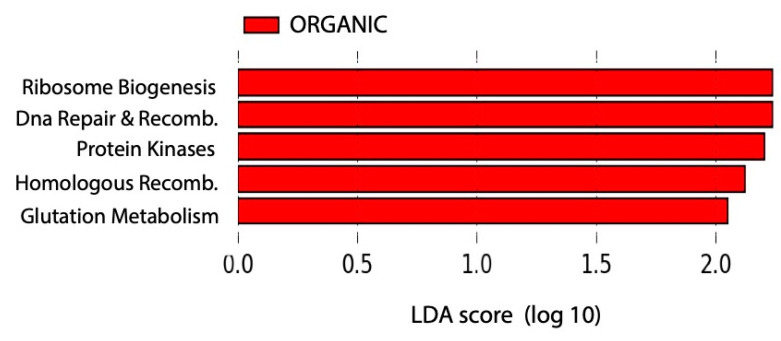
Predicted functions significantly differing across farming systems according to LEfSe analysis. LDA scores computed for PICRUSt predicted differentially abundant features (KEGG Orthologs L3) between conventional and organic soils. Histograms of the biomarkers identified between farming systems across sampling dates are available in Appendix A.

**Table 1 plants-09-01501-t001:** Effect of farming system on prokaryotes community structure. Microbial classes showing significant variation in their mean relative abundances by farming system (Kruskal–Wallis test, FDR *p* < 0.05). ^a^ indicates a higher value in conventional samples and ^b^ in organic samples.

Phyla	Class	Conventional Mean ± SD%	Organic Mean ± SD%
Firmicutes	Bacilli	1.62 ± 1.04 ^a^	0.68 ± 0.30
Acidobacteria	Solibacteres	1.11 ± 0.40 ^a^	0.69 ± 0.14
Planctomycetes	C6	0.06 ± 0.03 ^a^	0.02 ± 0.02
	Planctomycetia	1.71 ± 0.24 ^a^	1.39 ± 0.29
	vadinHA49	0.03 ± 0.02 ^a^	0.02 ± 0.01
Euryarchaeota	Methanomicrobia	0.00 ± 0.00	0.00 ± 0.01 ^b^
Proteobacteria	Deltaproteobacteria	8.77 ± 1.62	11.93 ± 2.85 ^b^
Chloroflexi	Anaerolineae	0.35 ± 0.08	1.36 ± 0.97 ^b^
Verrucomicrobia	Opitutae	0.20 ± 0.10	0.47 ± 0.20 ^b^
Fibrobacteres	Fibrobacteria	0.04 ± 0.05	0.07 ± 0.03 ^b^
Thermi	Deinococci	0.00 ± 0.00	0.01 ± 0.01 ^b^
Spirochaetes	Spirochaetes	0.00 ± 0.01	0.03 ± 0.04 ^b^
Bacteroidetes	Bacteroidia	0.00 ± 0.00	0.13 ± 0.20 ^b^
	Rhodothermi	0.00 ± 0.00	0.05 ± 0.04 ^b^
Chlorobi	Ignavibacteria	0.00 ± 0.00	0.01 ± 0.01 ^b^

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
