# Peer review of "Response of Horticultural Soil Microbiota to Different Fertilization Practices"

_plants, 2020, doi:10.3390/plants9111501_

Round 1
Reviewer 1 Report
Plants-964489: Response of horticultural soil microbiota to different fertilization practices (Zarraonaindia et. al.)
The current manuscript described how the application of chemical and organic (manure) fertilizers in farming practices affected soil physicochemical properties (e.g. pH, micronutrient levels), as well as soil microbiome richness, compositional structure, and shift in predicted functional pathways based on 16S rRNA gene metabarcoding data. The authors reported and discussed some agreement and discrepancy of the results from the current studies and those reported by early studies.
I find the manuscript being well-prepared, but I have issues with the experimental design and some missing information/results that can be interesting.
1) The authors described that the experiment was conducted in two separate plots (each under chemical or organic fertilization), but it was not clear if the tomato and swiss chard were cultivated in a mixture of in separate sub-plots (see 4.1. Field experimental design). It seems that there were no replicates/blocks for each experimental unit, e.g. sub-plots for each plant interspersed in space. Commonly we’d use split-design or RCBD design and to ensure at least 3 replicates under each treatment so statistical analysis would be meaningful. I acknowledge that the authors collected three soil samples per plant per sampling time point per fertilization practice, but you still need to have a defined experimental design strategy.
2) Another concern is that the amount of nutrient applied to the field under chemical and organic fertilization was not comparable. For example, if I understand correctly, the total amount of N in organic fertilizer was almost 9 times higher than that in chemical fertilizer applied in the field (Ln359-370). This alone should have a significant impact on the soil microbiome. The authors need to justify such a strategy and to explain if the shifts in soil microbiome were a consequence of so-called “organic farming” or simply the increase in nutrients supply.
3) I am surprised that the authors didn’t report on the yield/productivity of the crops, which could serve as a convincing indicator for beneficial farming practices. It would be interesting to know how the plants performed under chemical fertilization and manure.
4) I appreciate that the authors included two plants in the experimental design to conclude generalized effects under the two fertilization practices. However, it could still be interesting to know if organic farming imposed different impacts on different crops.
Minor issues:
L80-86 & 366-369: It is likely that organic fertilizers may introduce microbes to the fields
Ln85-86: The increase in soil microbiome diversity seems associated with the hosts’ growth stages.
Fig. 2D: No need to keep the last boxplot (“between” equals to “Conv. Vs. Org”)
Table 1. should be reported as MEAN±SD%
Ln195: Although you have discussed this at a later stage, it is worth noting here that functional prediction may not be accurate since the average bacterial OTUs in your samples were predicted from phylogenetically distant taxonomic lineages (79% identity).
Ln220-222: there are repeats of M&M and need to be removed.
Ln233-236: if chemical fertilizer causes soil acidification, the uptake of soil micronutrients can be significantly affected.
Ln244-263: what are the sources of the micronutrient elements (e.g. fertilizer formula, plant-microbe interaction?) What bacterial taxa may have contributed to the improved uptake of the micronutrients in the conv. and organ. fertilizing systems?
Author Response
Response to Reviewer 1 Comments
-Point 1) The authors described that the experiment was conducted in two separate plots (each under chemical or organic fertilization), but it was not clear if the tomato and swiss chard were cultivated in a mixture of in separate sub-plots (see 4.1. Field experimental design). It seems that there were no replicates/blocks for each experimental unit, e.g. sub-plots for each plant interspersed in space. Commonly we’d use split-design or RCBD design and to ensure at least 3 replicates under each treatment so statistical analysis would be meaningful. I acknowledge that the authors collected three soil samples per plant per sampling time point per fertilization practice, but you still need to have a defined experimental design strategy.
-Response 1: The experiment was conducted on two plots separated by 35 meters. At the starting point, both soils had same properties (same pH, CEC, OM content, OC, CaCO3, and N). Each of the plots (one intended for conventional, and the other for organic fertilization) were then divided into two subplots to plant tomato and Chards plants (one subplot per crop type). 3 replicated samples were collected in each subplot. We have tried to clarify it in the 4.1. Field experimental design section, and elsewhere in the text
In Field experimental design. Line 416-417: 25 days after, in June 2013, twenty-five seedlings of Swiss chard (Beta vulgaris subsp. adanensis) and tomato (Solanum lycopersicum) were transplanted into two sub-plots (one per crop type).
In results Line 101-105: A field experiment was conducted on adjacent plots to study the prokaryotes community diversity and composition of soils under different fertilization treatments, conventional vs. organic. In each of the plots two plant species (S. lycopersicum (tomato) and B. vulgaris (Swiss chard)) were planted in two subplots (one per crop type). Soils samples were collected over 3 months after fertilization and crop planting.
Point 2) Another concern is that the amount of nutrient applied to the field under chemical and organic fertilization was not comparable. For example, if I understand correctly, the total amount of N in organic fertilizer was almost 9 times higher than that in chemical fertilizer applied in the field (Ln359-370). This alone should have a significant impact on the soil microbiome. The authors need to justify such a strategy and to explain if the shifts in soil microbiome were a consequence of so-called “organic farming” or simply the increase in nutrients supply.
Response 2: The reason behind choosing the fertilizers used relies on the fact that both of them are commonly used in the conventional and organic farming systems in our territory, and we wanted to mimic the most common agricultural practices for the crops planted in our region to add value conclusions. The dosages used are also the commonly adopted ones to fulfill the necessity of the crops. The organic fertilizer is approved and certified by CAEE as ecological product; C qualification, and the dosages are the recommended ones. The microbial shifts are the respond of nutrient supply and their metabolization, which in this case, is linked to the different fertilizers used.
Point 3) I am surprised that the authors didn’t report on the yield/productivity of the crops, which could serve as a convincing indicator for beneficial farming practices. It would be interesting to know how the plants performed under chemical fertilization and manure.
Response 3: Authors agree this would have been a strength for the work. The study presented shows the first year analysis towards the understanding of the impact of farming system on microbial shifts and translocation of elements (in Liñero et al 2015, 2017) in horticultural plants. The plots under study are still being monitored and it is a parameter we are now taking into account.
Point 4) I appreciate that the authors included two plants in the experimental design to conclude generalized effects under the two fertilization practices. However, it could still be interesting to know if organic farming imposed different impacts on different crops.
Response 4: Chen et al. (2018) found that vegetation type (woody vs. herbaceous plants) was the most important variable (followed by pH) in explaining both bacterial and fungal soil community variations in a long-term farming system comparation. Therefore, one of our original hypotheses was that, as each crop type is expected to exude different metabolites into the rhizoplane, microbial differences would be found related to the crop type in the present study. However, we found no significant impact of crop type. Nor in alpha nor beta diversity. The later might be due to the combination of different factors: 1) The short term of the experiment, 2) the fact that we studied two herbaceous plants (tomato vs. Swiss chard), and 3) bulk soil microbiota was studied, rather than rhizosphere, and so the impact might have been under-detected.
-Minor issues:
-Point 5) L80-86 & 366-369: It is likely that organic fertilizers may introduce microbes to the fields
Response 5 : We agree that this is a fact and it is something that is commonly overlooked in the literature. Unfortunately, we did not analysed the microbial composition of the manure prior its application. We included n the text and added a reference
Line 255-258 While the higher richness in organic systems might be in part due to the introduction of microorganisms present in the manure into the soil (represented mainly by members within Firmicutes (Clostridia), Bacteroidetes and Chloroflexi) [22],
Line 328-330 Within Chloroflexi, members of the class Anaerolineae (SBR1031, SHA31) were enriched, which are known for their role in nitrogen cycling [6] and have been previously identified as a highly represented bacterial group in manure [22].
- St-Pierre, B.; Wright, A.-D.G. Comparative metagenomic analysis of bacterial populations in three full-scale mesophilic anaerobic manure digesters. Appl. Microbiol. Biotechnol. 2014, 98, 2709–2717, doi:10.1007/s00253-013-5220-3.
-Point 6: Ln85-86: The increase in soil microbiome diversity seems associated with the hosts’ growth stages.
Response 6: We modified the text to include this point, as we believe that the increase in soil diversity is due to both, the growth stage of the plants as well as the introduction of nutrients after fertilizers application.
Line 249-251 In the present study, that standardizes for differences in soil properties, crop type, and climate conditions, changes in soil microbial richness were observed over the duration of the experiment (3 months) associated with the crops developmental stage.
-Point 7: Fig. 2D: No need to keep the last boxplot (“between” equals to “Conv. Vs. Org”)
Response 7: The ” between” boxplot has been removed.
-Point 8: Table 1. should be reported as MEAN±SD%
Response 8: The table has been changed accordingly
-Point 9: Ln195: Although you have discussed this at a later stage, it is worth noting here that functional prediction may not be accurate since the average bacterial OTUs in your samples were predicted from phylogenetically distant taxonomic lineages (79% identity).
Response 9: Added
Line 223-224 Using PICRUSt, the proportions of functional genes for each community were predicted (Figure S5) for the sequences that had a hit with the Greengenes reference OTUs at >97% identity.
-Point 10: Ln220-222: there are repeats of M&M and need to be removed.
Response 10: This sentence has been removed as suggested.
-Point 11 Ln233-236: if chemical fertilizer causes soil acidification, the uptake of soil micronutrients can be significantly affected.
Response 11: This statement has been included and a reference has been added.
Line 264-265: pH is known to influence microbial composition [25–29] as well as the mobility of heavy metals, influencing micronutrients uptake [30]
- Yang, S.-X.; Liao, B.; Li, J.; Guo, T.; Shu, W.-S. Acidification, heavy metal mobility and nutrient accumulation in the soil–plant system of a revegetated acid mine wasteland. Chemosphere 2010, 80, 852–859, doi:10.1016/j.chemosphere.2010.05.055.
-Point 12 :Ln244-263: what are the sources of the micronutrient elements (e.g. fertilizer formula, plant-microbe interaction?) What bacterial taxa may have contributed to the improved uptake of the micronutrients in the conv. and organ. fertilizing systems?
Response 12. The experiment was conducted on a plot that was divided into two, one intended for organic the other intended for conventional farming. Their soil was the same at the starting point (there was no significant difference among the soil edaphic properties of the plots prior fertilizer amendment and plant plantation). Therefore the micronutrient differences between the plots would have it source in the fertilizer formula at first, however their mobility and bioavailability and their accumulation in the plant would be a result of the of fertilizers changing soil edaphic properties ( for instance changes in pH, Total C etc), microbes metabolic activity as well as plant -microbe interaction through root exudates.

Reviewer 2 Report
The paper reports the effect of different fertilization practices on the soil populations of prokayotes over a three months period after a single application of two different fertilizers. A chemical fertilizer NPK and an organic fertilizer (based on certified horse manure) were applied on adjacent plots with the same land use history. Two different phytosanitary strategies were also adopted: insecticides and fungicides were applied in the chemically fertilized plot and Tagetes was planted in the borders of the "organic plot" as natural repellent together with copper sulfate applications, admitted in organic farming against fungi. Two different crops were planted soon after fertilization, Tomato and Beta vulgaris. The results are based on 16S rRNA gene amplicon sequencing of DNA extracted from soil sampled in three campaigns on a period of three months after seeding (june-august). The results show that the prokaryotic communities of the fertilized plot are different from those of the organic plot with those from the first sampling soon after fertilization being the most differentiated one. Functions and role of the differentially abundant taxa under the two soil management are inferred by PICRUSt and discussed.
The manuscript is well written, generally clear and in good English. The experimental design is correct. Figures and tables are clear and well explained. The main merit of the paper is that the field research is based on a standardized field experiment where the two differentially managed plots share all other characteristics such as soil type, land use history, environment etc. This introduces less variables and makes comparisons more reliable.
The main concern, conversely, is related to the short term experimental period that is quite uncommon in agriculture and in general in field contexts. Monitoring of the effects is only carried out in three months, immediately after a single different fertilization treatment. The shift in the prokaryotic populations affected by the two different fertilizers could be just transient and of low significance in the long term. Also an important soil parameter, pH is only temporarily affected by the fertilization treatment and will probably come back to the previous value in a longer period. The results of this article does not answer to important questions on the long term effect of soil management on C accumulation and other soil ecosystem services. In long term experiments the differences in bacterial populations as affected by different soil management are often less evident (see for example Novara et al., Sustainability 2020, 12, 3256; doi:10.3390/su12083256). However the results of this paper suggest the trends of bacterial population changes that could be monitored and confirmed in longer experimental monitoring.
If the authors have this limit clear, they should present it in the introduction and consequently discuss it in the discussion section. Also conclusions should report that further research is needed to confirm these results in the long period.
A second comment regards the definition of farming system (organic system vs. conventional system) that is used in the text in a too simplistic meaning. An organic system defines the whole production and management system that is applied in agriculture that is much more than one single organic fertilization. Organic farming aims at sustainability by reducing external inputs and requires a deep revolution of the soil management conception that includes conservative soil tillage, improvement of biodiversity, organic matter increase etc. Although the term farming system can be used to “label” the two different soil plots it should be presented in its full meaning at least in the introduction.
Specific comments
Line 20 eliminate an application of
Lines 20-23 unclear sentence. Please reformulate
Results. Due to the style of the journal, results that come soon after the introduction need a brief summary of the experimental design to be clear enough.
Line 207. Overexpressed is not correct. PICRUST predicts functions but not gene expressions. I suggest over-represented.
Line 261. Cu is also added as copper sulfate to the plants as fungicide in the organic fertilized plot
line 320. Rhodococcus
Line 391. what was the DNA yeld extracted from each soil ? was organic soil richer in DNA?
Author Response
Response to Reviewer 2 Comments
-Point 1: The main concern, conversely, is related to the short term experimental period that is quite uncommon in agriculture and in general in field contexts. The results of this article does not answer to important questions on the long term effect of soil management on C accumulation and other soil ecosystem services. In long term experiments the differences in bacterial populations as affected by different soil management are often less evident (see for example Novara et al., Sustainability 2020, 12, 3256; doi:10.3390/su12083256). However the results of this paper suggest the trends of bacterial population changes that could be monitored and confirmed in longer experimental monitoring.
If the authors have this limit clear, they should present it in the introduction and consequently discuss it in the discussion section. Also conclusions should report that further research is needed to confirm these results in the long period.
Response 1: We agree that the short-term of the experiment is a limitation of the study, and included it in the introduction, discussion and conclusion as suggested. The plots are still being monitored and we aim to collect samples again to confirm the results in the longer term. However, we thought that the results presented here were interesting enough for publication, as a first approach, as conclusions were comparable to longer term studies, it had a multidisciplinary approach (correlating microbes with plant element accumulation), and the experimental design controls for various confounding factors ( crop type, soil properties etc).
in the introduction Line 85: Extending the duration of the experiment would help to resolve whether the microbial shifts observed here are persistent over time and determine if the treatments have an impact on soil quality on the long-term, that is ultimately required for evaluating sustainability of land-use regimes.
in discussion Line 310-314: However, the experiment conducted should be extended on time to assess whether the microbial shifts associated to the farming system and the differential uptake of elements by the crops under study persist, in order to evaluate farming system’s impact on the quality and health of soil, and hence, the sustenance of the system.
in conclusions Line 533-535 However, further research is needed to confirm these results in the long-term, that altogether would allow elucidating the connection between the observed changes and plant productivity, disease resistance and stress resilience.
-Point 2: A second comment regards the definition of farming system (organic system vs. conventional system) that is used in the text in a too simplistic meaning. An organic system defines the whole production and management system that is applied in agriculture that is much more than one single organic fertilization. Organic farming aims at sustainability by reducing external inputs and requires a deep revolution of the soil management conception that includes conservative soil tillage, improvement of biodiversity, organic matter increase etc. Although the term farming system can be used to “label” the two different soil plots it should be presented in its full meaning at least in the introduction.
Response 2: Totally agree. We have revised the text to reflect a broader context and reflected that we will be referring to our samples as organic and conventional farming for simplicity.
Line 41: Organic systems, defined by management practices lacking the application of synthetic fertilizers and pesticides, appears to reduce the burden of xenobiotics in the food chain [3],
Line 51-60: Organic fertilizers are known to have several advantages to improve soil fertility, such as the ability to increase organic matter content in soil, improve the soil structure, enhancing soil nitrogen content, nutrient availability, improving nutrient mobilization as well as increasing root growth [10]. Organic practices rely upon crop rotations, crop residues, animal and/or green manure, off-farm organic wastes, mechanical cultivation, mineral bearing rocks, and aspects of biological pest control to maintain soil productivity and supply plant nutrients.
10 Kumar Bhatt, M.; Labanya, R.; Joshi, H.C. Influence of Long-term Chemical fertilizers and Organic Manures on Soil Fertility - A Review. Univers. J. Agric. Res. 2019, 7, 177–188, doi:10.13189/ujar.2019.070502.
Line 117: (hereafter referred as conventional and organic farming for simplicity).
-Point 3: -Line 20 eliminate an application of
Response 3. Done
-Point 4: -Lines 20-23 unclear sentence. Please reformulate
Response 4. Done
Line 20: 16S rRNA sequencing was applied to soils from adjacent plots receiving either a synthetic or organic fertilizer, where two crops were grown within treatment, homogenizing for differences in soil properties, crop and climate
-Point 5: Results. Due to the style of the journal, results that come soon after the introduction need a brief summary of the experimental design to be clear enough.
Response 5: The following sentence has been added
Line 101-105: A field experiment was conducted on adjacent plots to study the prokaryotes community diversity and composition of soils under different fertilization treatments, conventional vs. organic. In each of the plots two plant species (S. lycopersicum (tomato) and B. vulgaris (Swiss chard)) were planted in two subplots (one per crop type). Soils samples were collected over 3 months after fertilization and crop planting.
-Point 6: Line 207. Overexpressed is not correct. PICRUST predicts functions but not gene expressions. I suggest over-represented.
Response 6: Agree. Overexpressed has been changed for “Over-represented”
-Point 7: Line 261. Cu is also added as copper sulfate to the plants as fungicide in the organic fertilized plot
Response 7: True, it has been included
Line 304-305:For instance, due to the spraying of copper sulfate on plant aerial parts, a higher accumulation of Cu might be expected in organic roots. Besides, its higher concentration in organic agricultural practices have been previously associated to a higher presence of arbuscular mycorrhizal fungi (AMF)
-Point 8: Line 320. Rhodococcus
Response 8: Changed
-Point 9: Line 391. what was the DNA yield extracted from each soil ? was organic soil richer in DNA?
Response 9: The total genomic DNA varied considerably between samples, but did not show any association with the farming type, likely due to the short term of the experiment and the soils just been fertilized ones. That would have been a very interesting result to find.
